# Nuclear mechano-confinement induces geometry-dependent HP1α condensate alterations
Oda Hovet [1], Negar Nahali[2], Andrea Halaburkova[1], Linda Hofstad Haugen[1], Jonas Paulsen [1,2] ✉ & Cinzia Progida [1] ✉

Cells sense external physical cues through complex processes involving signaling pathways, cytoskeletal dynamics, and transcriptional regulation to coordinate a cellular response. A key emerging principle underlying such mechanoresponses is the interplay between nuclear morphology, chromatin organization, and the dynamic behavior of nuclear bodies such as HP1α condensates. Here, applying Airyscan super-resolution live cell imaging, we report a hitherto undescribed level of mechanoresponse triggered by cell confinement below their resting nuclear diameter, which elicits changes in the number, size and dynamics of HP1α nuclear condensates. Utilizing biophysical polymer models, we observe radial redistribution of HP1α condensates within the nucleus, influenced by changes in nuclear geometry. These insights shed new light on the complex relationship between external forces and changes in nuclear shape and chromatin organization in cell mechanoreception.

Cells constantly sense and respond to mechanical inputs, and the interplay between nuclear deformation, morphology and chromatin organization emerges as a vital process underlying cell mechanoresponse[1,2]. Recent studies show that the nucleus can act as a "ruler" to measure cellular and nuclear shape variations originating from external compression, to interpret and respond to cues important for survival, movement and growth[3,4].

Within the nucleus, chromatin is increasingly recognized as a prominent mechanosensor capable of translating mechanical inputs into transient or lasting alterations in gene expression[2,5–12]. For example, cyclic stretching of cells resulted in loss of heterochromatin and induced nuclear softening[13]. Moreover, mechanical compression of fibroblasts increased heterochromatin levels through the import of histone deacetylase 3 (HDAC3) into the nucleus[14]. During confined 3D migration, global heterochromatin foci was observed to be persistently formed and lasting from hours to days in cancer cells and fibroblasts, coupled with reduced transcriptional activity[5].

Heterochromatin formation and maintenance involve interaction with proteins in the heterochromatin protein 1 (HP1) family[15,16]. HP1α binds specifically to H3K9me2,3-marks upon homodimerization to form liquid-liquid phase separation condensates through weak hydrophobic interactions[17–22]. HP1α condensates contribute to 3D genome organization and bring distal heterochromatic regions together in response to external cues[17–22].

HP1α proteins exhibit dynamic behavior on the scale of seconds and arrange larger DNA molecules into condensed, long-lasting domains over

the course of hours[21,23], possibly through increasing the effective viscosity to maintain stable condensate structures[24]. However, HP1α condensates can be rapidly disassembled in response to environmental and developmental cues[21]. In addition, HP1α condensates resist high instantaneous disruptive forces of at least 40 pN and are able to further resist transient external force[24]. This suggests that HP1α-driven heterochromatin formation plays a critical role in mechanoresponse to external stimuli, yet, how mechanical alteration of the nucleus translates to specific responses in HP1α remains elusive.

In this study, we investigate the interplay between cell confinement, nuclear shape and HP1α condensates in mechanotransduction. To achieve this, we apply force to cells with controlled micrometer precision in a static multi-well confiner, thereby confining cells from 2 to 5 μm heights, facilitating imaging across several conditions. Through a combination of experimental and modeling approaches, we characterize 3D geometric and morphological changes of the cell nucleus and HP1α condensates subjected to nuclear confinement below their resting nucleus diameter. In response to nuclear confinement, live cell imaging in human lung fibroblasts and cervical cancer cells shows increased nuclear flattening and subsequent reduction in the number and size of HP1α condensates, as well as reduced mobility. Combining biophysical polymer modeling with imaging data reveals a centralization of HP1α condensates upon increased nuclear flattening, suggesting that nuclear geometrical constraints substantially impact heterochromatin localization. Taken together, these findings contribute to

[1]Department of Biosciences, Faculty of Mathematics and Natural Sciences, University of Oslo, Oslo, Norway. [2]Centre for Bioinformatics, Department of Informatics, University of Oslo, Oslo, Norway. ✉e-mail: jonas.paulsen@ibv.uio.no; c.a.m.progida@ibv.uio.no

our understanding of how the nucleus and the formation and disintegration of heterochromatic domains participate in mechanotransduction processes.

## Results

### Cell confinement induces nuclear flattening

To conduct an in-depth examination of the nuclear effects brought upon by specific compressive forces to the cell, we implemented a system allowing cell confinement at micrometer precision (Fig. 1A)[25–27]. Fetal lung fibroblast IMR90 or HeLa cells expressing HP1α-GFP were imaged by Airyscan super-resolution microscopy. We computed nuclear isosurfaces from live-cell measurements of HP1α-GFP to measure the effect of ~15 min cell confinement on nuclear morphology (Fig. 1B, E). We measured the resting nuclear diameter of IMR90 cells to a median of 5.99 micrometers (Fig. 1C). Confining cells at or below this resting diameter is expected to affect nuclear morphology. We thus confined IMR90 cells to 2–5 μm, and measured a range of parameters describing geometric features of the nucleus. We first confirmed the effect of cell confinement by measuring the height of the nuclei, which resulted in a median height of 3.79 μm (−37% from non-confined) when cells were confined between 3–5 μm, and 2.57 μm (−57% from non-confined) when confined below 3 μm (Fig. 1C). We then measured nuclear flatness, calculated as the square root of the ratio of the minimal and major axes (see Methods). Using this approach, we observed a significant increase in nuclear flatness from an initial median score of 0.48 in non-confined cells to 0.58 at <3 μm confinement (Fig. 1D). Thus, our cell confinement system induces specific nuclear flattening and morphological change in IMR90 nuclei.

To validate our confinement system in a cancer cell line, we repeated our analyses in human cervical cancer cells (HeLa). As these cells have a larger resting nuclear height (median 10.06 μm, Fig. 1E, F), we confined them to 5–8 μm. In these conditions, we did not observe any significant volume change, as reported earlier by Lomakin et al.[3]. However, higher degree of confinement (<5 μm) resulted in a relatively similar degree of confinement (~50%) compared to IMR90 cells (Supplementary Fig. 2), which is in line with micropipette aspiration experiments previously reported by Rowat et. al.[28]. As for IMR90 cells, cell confinement induces a reduction in the minor (confinement) axis of the cells, from 10.06 μm in non-confined cells to 5.49 μm (−45% from non-confined) at 5–8 μm confinement, and 4.33 μm (−57% from non-confined) at <5 μm confinement (Fig. 1F). Finally, nuclear flatness increases significantly from 0.27 in non-confined cells to 0.37 at 5–8 μm confinement, and to 0.42 μm at <5 μm confinement (Fig. 1G). While sphericity of nuclei is reduced in confined cells, the absence of observed elongation (Supplementary Fig. 1) can be explained by the reduction in all axes resulting in a decrease in nuclear volume in IMR90 and HeLa (Supplementary Fig. 2C, F). As elongation is determined based on intermediate and major axes, alterations in these dimensions influence the calculated ratio (see Methods) (Supplementary Fig. 2A, B, D, E). We conclude that our cell confinement system induces nuclear flattening to similar degrees between the confinement states in IMR90 and HeLa.

### Confined nuclei show a reduction in the number and mobility of HP1α condensates

Having established a reliable and robust nuclear confinement system, our objective was to investigate the potential impact of confined nuclear geometry on chromatin structure. Heterochromatin protein alpha (HP1α) is a well-studied protein known to form phase-separating condensates in human nuclei linked to heterochromatin formation and maintenance[17,18,24]. By overexpressing HP1α-GFP, we confirm a median number of ~12 HP1α condensates per nucleus in IMR90 and ~10 in HeLa cells (Fig. 2A–C), with a median size of ~780 nm in IMR90 and ~624 nm in HeLa (Fig. 2D). The HP1α condensates colocalize with H3K9me3 with a median of 70% in IMR90 and 77% in HeLa, confirming the heterochromatic nature of the observed condensates (Fig. 2E). Colocalization is also confirmed upon confinement of HP1α-GFP with H3K9me3 in IMR90 and HeLa (Supplementary Fig. 3). HP1α condensate properties in IMR90 and HeLa cells are in

line with dimensions reported in other cell lines[29], thereby confirming these as reliable indicators of heterochromatin properties in our confinement system.

To investigate the response of HP1α to rapid confinement, we repeated the same measurements on cells confined for ~15 min. We observe a reduction from a median of ~16 condensates to ~8 condensates in confined nuclei (−50%) from live-cell super resolution imaging of HP1α-GFP in IMR90 (Fig. 3A, B). A similar reduction upon confinement is seen in live HeLa cells, from ~11 condensates to ~6 condensates (−45%) (Fig. 3E, F). A reduction is observed also for endogenous HP1α (from 17 to 6 condensates resulting in a reduction of 65% in IMR90, Supplementary Fig. 4A, B), indicating that the overexpression of the protein does not considerably affect the response to confinement. Moreover, the reduction in number of HP1α condensates in confinement is not due to a decrease in total protein expression as shown in Western blot analysis (Supplementary Fig. 4D, E), nor to the decreased nuclear volume (Fig. 3D and H). The diameter sizes of the condensates significantly decrease upon confinement, in HeLa (−24%) and IMR90 (−13%) (Fig. 3C and G) which is also observed for endogenous HP1α in IMR90 (−18%) (Supplementary Fig. 4C). The number of detected condensates and size measurements vary across non-confined and confined states due to different cell lines and factors that influence condensate structures such as phosphorylation, differences in HP1α, protein concentrations and molecular interactions with other proteins[24,30]. The condensate decrease seems to be specific for HP1α and not due to general chromatin redistribution, as quantification of intense DAPI signal (corresponding to strong chromatin foci) does not reveal any difference in the number of chromatin foci between confined and not confined cells, and only a moderate decrease (−11%) of their diameter upon confinement (Supplementary Fig. 5A–C). The fact that only HP1α condensates are specifically affected is in line with previous work demonstrating that displacement of HP1 proteins from pericentromeric heterochromatin does not result in changes in DAPI-stained heterochromatin domains[31]. In conclusion, HP1α condensates are reduced in number and size upon nuclear confinement with median reductions between 45–65% and 13–24%, respectively.

The reduction in condensate numbers is observed immediately after confinement (within ~15 min) and is reverted upon confinement release (Supplementary Fig. 5D). During confinement, each condensate generally displays limited movement with mean squared displacements (MSD) showing a slow trend over time (Fig. 4A and Supplementary video 1). By plotting the log of mean MSD over time and fitting to the first ~7 h, where data abundance is highest, we observed that both non-confined and confined cells exhibit condensates with subdiffusive behavior with an anomalous diffusion exponent (α) of 0.5 and 0.8, respectively (Fig. 4A, B). This indicates very stable positioning of condensates typical for heterochromatic particles in the nucleus[32,33] (Fig. 4C, D). The observed reduction in number of condensates is not due to a reduced half-life of condensates in confined cells, as condensate lifetime is significantly increased in confined cells compared to non-confined cells (Supplementary Fig. 6B). We conclude that HP1α condensates generally exhibit subdiffusive behavior. Upon confinement, condensate dynamics is slower, and condensates are reduced in number (~−45–65%) and in size (~−13–24%).

To assess whether confinement in addition to slowing down condensate dynamics also affects the exchange dynamics of HP1α, we performed Fluorescence recovery after photobleaching (FRAP) experiments on HP1α-GFP in HeLa cells, comparing the fluorescence recovery within bleached regions inside and outside condensates in control and confined conditions. The results indicate that HP1α is still able to exchange dynamically in confined cells, even though the exchange kinetics are significantly slower than in non-confined cells (Fig. 5). However, while the fluorescence recovery reached a maximum of circa 90% for ROIs bleached either within or outside condensates in non-confined cells, in confined cells HP1α-GFP fluorescence recovery reached only 70% as a maximum in regions outside the condensates and less than 60% within the condensates. This may suggest that the fraction of HP1α bound to chromatin is more stably associated upon confinement.

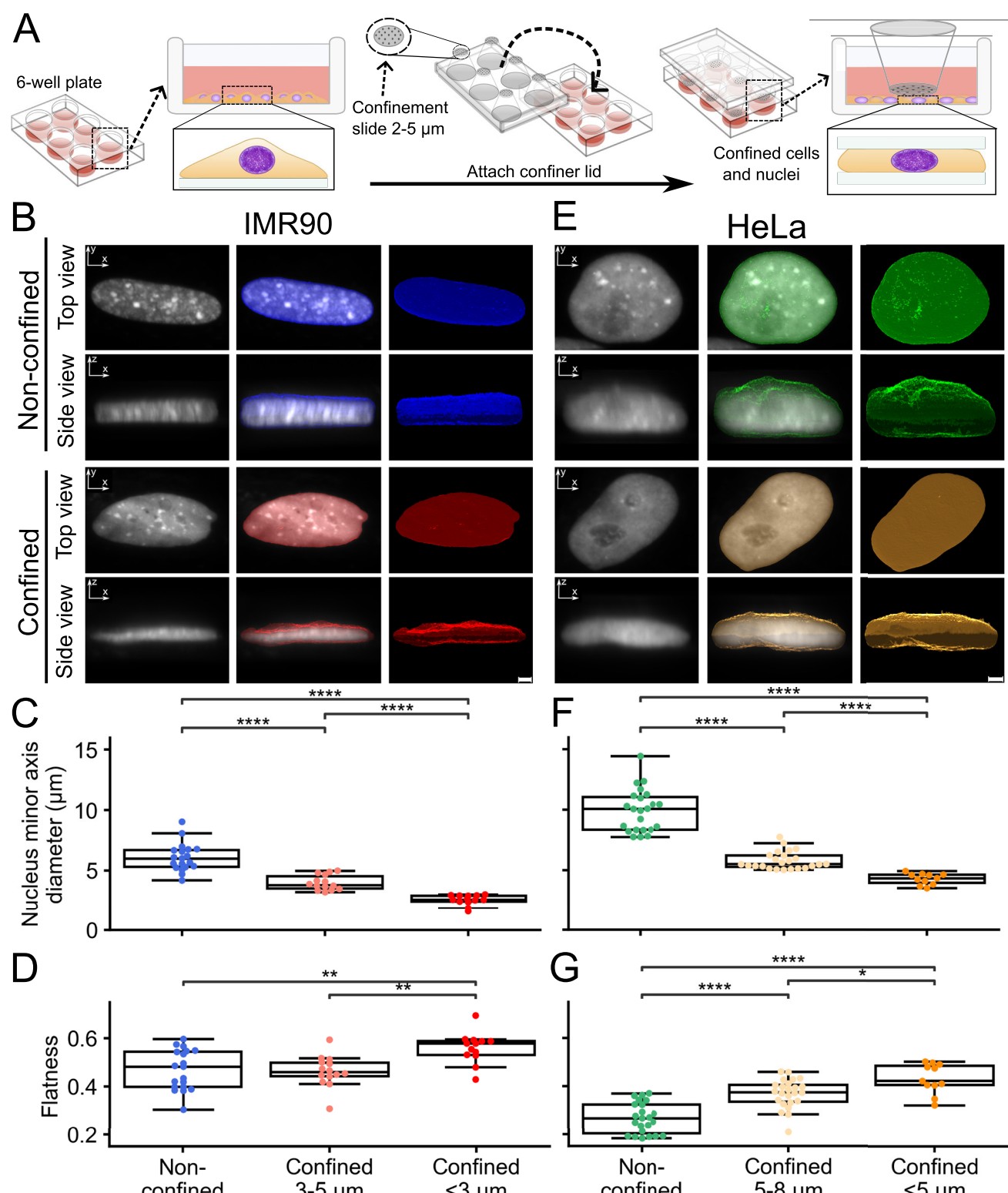

**Fig. 1 | Nuclear properties upon static cell confinement. A** Overview of the static cell confinement system (3 µm confinement for IMR90 and 5 µm for HeLa cells). **B** Representative images of non-confined and confined nuclei for IMR-90 cells shown in raw (black and white) and isosurface (coloured) top- and side views. Scale bar: 2 µm. **C** Boxplots showing height (in µm) in non-confined (left box, blue), confined to 3–5 µm (middle box, pink) and below 3 µm (right box, red) in IMR90 cells (19 in non-confined, 14 in 3–5 µm and 13 in <3 µm, from 6 independent experiments). **D** Boxplots showing the measured flatness of the same nuclei as in (**C**) (flatness of 1 indicates a 1-dimensional flat plane). **E** Example images of non-

confined and confined nuclei for HeLa cells shown in raw and isosurface top- and side views. Scale bar: 2 µm. **F** Boxplots showing height (in µm) in non-confined (left box, green), confined to 5–8 µm (middle box, yellow) and below 5 µm (right box, orange) in HeLa cells (23 in non-confined, 24 in 5–8 µm and 11 in <5 µm, from 3 independent experiments). **G** Boxplots showing the measured flatness of the same nuclei as in panel F. The boxplots in (**C**, **D**, **F**, **G**) indicate median (middle line), 25th, 75th percentile (box) and largest and smallest values extending no further than 1.5 × interquartile range (whiskers), except for outliers. *: 1.00e-02 < $p$ ≤ 5.00e-02, **: 1.00e-03 < $p$ ≤ 1.00e-02, ****: $p$ ≤ 1.00e-04 from Mann–Whitney test two-sided.

**Fig. 2 | HP1α condensate properties in resting conditions in IMR90 and HeLa. A** Airyscan super resolution images of nuclei IMR90 transiently transfected with HP1α-GFP (green), fixed and stained with anti-H3K9me3 (red) antibodies. Merged image shown in the right panel. Scale bar = 2 μm. **B** HeLa cells stably transfected with HP1α-GFP (green), fixed and stained with antibody against H3K9me3 (red). Merged image shown in the right panel. Scale bar = 2 μm. **C** Boxplots showing number of HP1α condensates in IMR90 and HeLa (18 in IMR90 and 32 in HeLa, from 3 independent experiments each). **D** Boxplots showing diameter (in μm) of individual HP1α condensates in IMR90 and HeLa (258 condensates in IMR90 and 327 in HeLa). **E** Co-localization analysis of HP1α vs. H3K9me3 with Mander's coefficient to indicate overlap. The boxplots in (**C–E**) indicate median (middle line), 25th, 75th percentile (box) and largest and smallest values extending no further than 1.5 × interquartile range (whiskers), except for outliers.

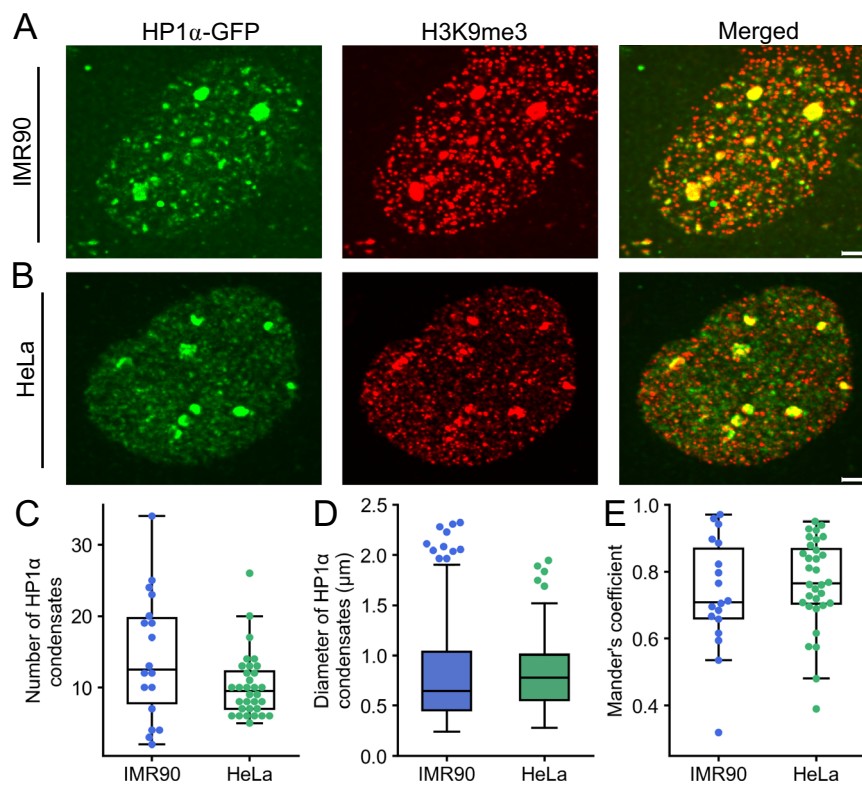

## Nuclear confinement induces nuclear geometry-dependent redistribution of HP1α condensates

To next investigate if nuclear HP1α condensates redistribute in the nucleus upon confinement, we computed the distance from the center of each condensate to the nuclear center and minimum distance to the nuclear periphery in individual isosurface-rendered nuclei from IMR90 cells expressing HP1α-GFP (Fig. 6A). We contrasted these distances to expected distances generated by randomly sampling condensate control positions in spaces defined by the radius of each condensate within the same nuclei (Fig. 6B, C; dotted lines) (see Methods). Comparing expected and observed distances to the nuclear center, we note relatively similar distributions for non-confined HP1α condensates, whereas a less similar distribution for confined HP1α condensates (Fig. 6B). However, observed HP1α condensate distances to the nuclear periphery are strikingly different to expected distances (Fig. 6C), indicating that HP1α condensates are more centrally placed than expected by chance. When comparing distributions of distances between confined and non-confined nuclei, we note a tendency of observed HP1α condensate to be distributed even more towards the nuclear center (Fig. 6B), and away from the nuclear periphery (Fig. 6C) upon confinement. When comparing distributions for degrees of confinement, we confirm a statistically significant shift of HP1α towards the nuclear center (Fig. 6D) and away from the nuclear periphery (Fig. 6E). Thus, in confined IMR90 nuclei, HP1α condensates are radially redistributed. This effect is not seen in HeLa cells suggesting that this is a cell-type dependent response (Supplementary Fig. 7).

## Polymer modeling suggests geometry-dependent HP1α condensate repositioning upon nuclear confinement

The redistribution of HP1α condensates towards the nuclear center may appear counterintuitive since a flattened nucleus would theoretically have less space in the center, thereby reducing the likelihood of observing condensates in that region. To explore the underlying biophysical principles behind the surprising centralization of HP1α condensates under confinement, we utilized polymer models incorporating H3K9me3-driven HP1α-crosslinking, taking into account spherical and flattened nuclear geometries

(Fig. 7A, B) (see Methods). To this end, we embedded polymer models inside shells whose sizes were chosen so that the fraction of the volume occupied by the monomers is approximately 10%, mimicking the nuclear fraction occupied by chromatin in human cells[34].

The shape of the nucleus shells were varied from spherical to oblate (or flattened) shapes, mimicking nuclear flattening seen upon cell confinement. To explore the effects in more detail, we considered two distinct oblate cases. In one, the minor axis was half of the major axis, and in the other, it was a third of the major axis.

In order to model the interactions between HP1α proteins and chromosomes (polymers), we utilized Human IMR90 H3K9me3 ChIP-seq data to extract relative values of the interaction strengths used to determine the interaction parameter (ε) for the Lennard-Jones potential. To identify HP1α condensates in the models, we employed agglomerative hierarchical clustering with single linkage on the monomers (see Methods for details).

Modeling using low interaction strength (E1) between HP1α binding sites (H3K9me3 regions) (Fig. 7C), reveals that flattened nuclei (red and green lines) indeed display HP1α-condensates with shorter average distance to the nuclear center than spherical nuclei (red line). Increasing the interaction strength (ε) enhances this difference further (Fig. 7E).

Furthermore, the extent of nucleus flattening also appears to play a role, as flatter nuclear models (red lines) generally display more pronounced deviation from the spherical models (blue) compared to less flattened ones (green lines).

Similarly, comparing distances of modeled HP1α condensates to the nuclear periphery, shows that distances slightly increase in flattened relative to spherical nuclei (Fig. 7D), however, the difference is not enhanced upon an increase in HP1α binding strength (Fig. 7F).

This modeled redistribution of HP1α condensates in flattened nuclei is in line with the centralization of condensates in confined nuclei as measured for HP1α condensates in IMR90 (Fig. 6).

Similar modelling however reveals that the number of HP1α condensates is independent of the shape of the confinement (Supplementary Fig. 8). We thus conclude that the observed centralization of HP1α condensates, but not their number, in IMR90 is influenced by the change of

**Fig. 3 | HP1α condensates are reduced in number and sizes upon immediate (~15 min) confinement. A** Representative isosurface-rendered images of non-confined and confined (3 μm) nuclei in live IMR90 cells transiently transfected with HP1α-GFP shown in top- and side views. **B** Boxplots showing number of HP1α condensates in IMR90 cells (19 in non-confined, 14 in 3–5 μm and 13 in <3 μm, from 6 independent experiments). **C** Boxplots showing diameter sizes (in μm) of individual HP1α condensates in IMR90 cells (left box; 261 condensates [in 19 nuclei] in non-confined, middle box; n = 98 [in 14 nuclei] in 3–5 μm and right box; n = 83 [in 13 nuclei] in <3 μm). **D** Boxplots showing total HP1α condensate volumes scaled by respective nuclei volumes in IMR90 cells (19 in non-confined, 14 in 3–5 μm and 13 in <3 μm, from 6 independent experiments). **E** Representative isosurface-rendered images of non-confined and confined (5 μm) nuclei in HeLa cells stably expressing HP1α-GFP shown in top- and side views. **F** Boxplots showing number of HP1α condensates in HeLa cells (23 in non-confined, 24 in 5–8 μm and 11 in <5 μm, from 3 independent experiments). **G** Boxplots showing diameter sizes (in μm) of individual HP1α condensates in HeLa cells (left box; n = 224 condensates [in 23 nuclei] in non-confined, middle box; n = 170 [in 24 nuclei] in 5–8 μm and right box; n = 74 [in 11 nuclei] in <5 μm). **H** Boxplots showing total HP1α condensate volumes scaled by respective nuclei volumes in HeLa cells (23 in non-confined, 24 in 5–8 μm and 11 in <5 μm, from 3 independent experiments). The boxplots in (**B–D** and **F–H**) indicate median (middle line), 25th, 75th percentile (box) and largest and smallest values extending no further than 1.5 × interquartile range (whiskers), except for outliers.*: 1.00e-02 < p ≤ 5.00e-02, **: 1.00e-03 < p ≤ 1.00e-02, ***: 1.00e-04 < p ≤ 1.00e-03, ****: p ≤ 1.00e-04 from Mann–Whitney test two-sided.

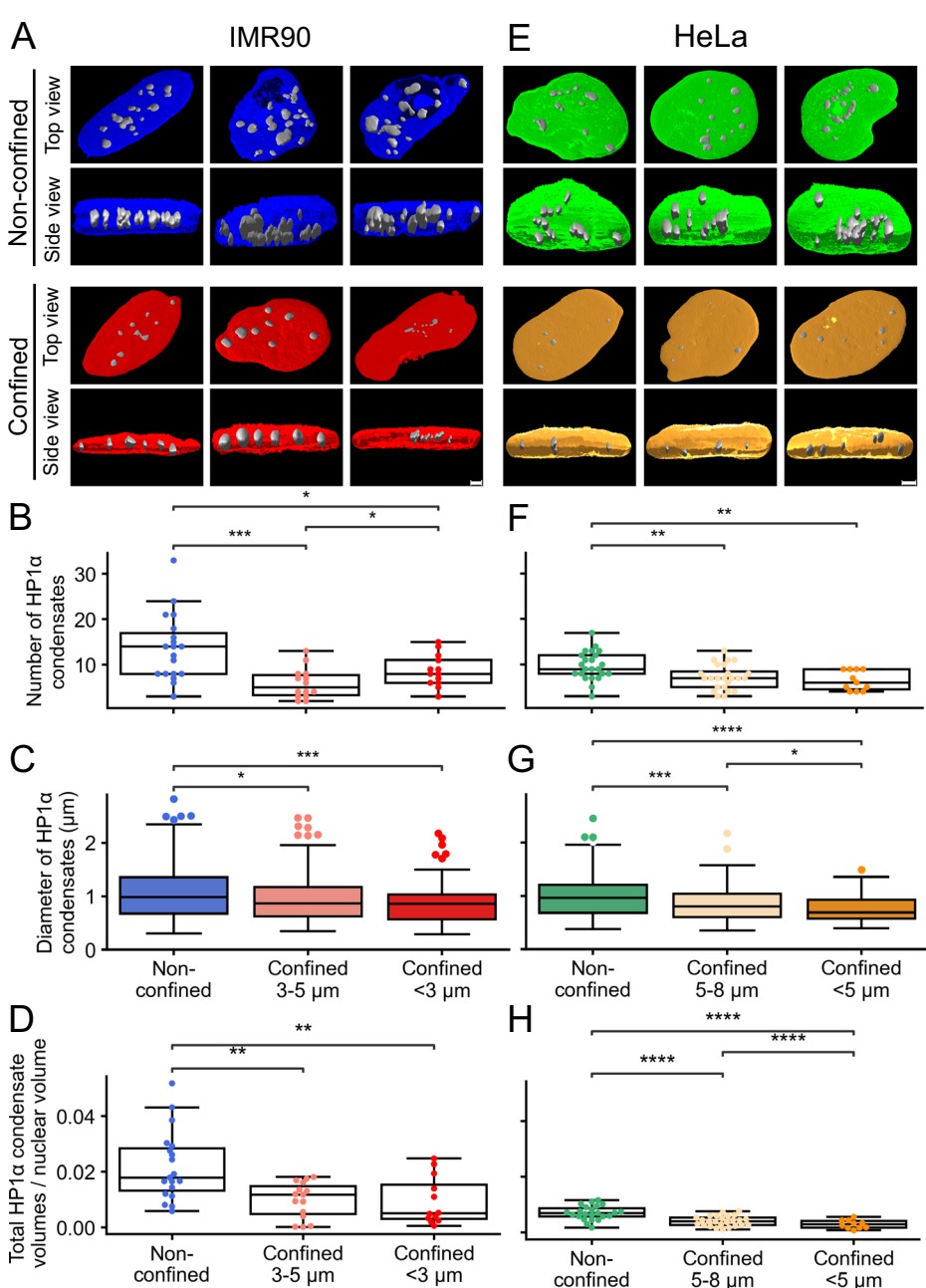

geometric shape of the nucleus, and is emphasized by the binding strength of HP1α, suggesting centralization is entropically favorable when the nucleus is more confined.

## Discussion

Cells constantly sense and respond to physical cues from their surrounding environment, with the nucleus emerging as a key mechanoresponder. Here, we apply a micrometer precision cell confinement system coupled with biophysical modeling to study mechanoresponse at nuclear and chromatin levels. Our cell confinement system systematically induces nuclear flattening (Fig. 1B–G), which in turn results in a (~ −45–65%) reduction in HP1α condensates (Fig. 3), reduced condensate size (~−13–24%), reduced dynamics (Fig. 4) and HP1α condensates repositioning to the nuclear center (Fig. 6). Biophysical modeling of HP1α condensate formation suggests that flattened nuclei induces condensate nuclear centralization (Fig. 7).

In line with our observations, a reduction in the number of HP1α condensates has also been reported in mouse fibroblasts upon application of static compressive forces to cells[14]. Furthermore, condensates in confined states experience force-dependent disintegration[24], which possibly explains their reduced numbers and decreased diameter sizes observed in live cell and immunofluorescence imaging (Fig. 3 and Supplementary Fig. 4) across IMR90 and HeLa cells. The exact mechanisms involved in this disintegration remain unresolved. However, HP1α condensate formation and disassembly is known to depend on an intricate interplay between HP1α-HP1α and HP1α-DNA binding kinetics, HP1 inter-paralog competition, DNA viscosity and forces applied directly to condensates[24]. Studies have shown that mechanical inputs reduce peripheral H3K9me3 heterochromatin[13,35], potentially explaining our observed decrease and centralization of HP1α condensates upon confinement, which is known to be affected by H3K9me3 levels[36]. The reduced condensate dynamics and exchange kinetics we observe (Figs. 4 and 5), on the other hand, could potentially be attributed to

**Fig. 4 | HP1α condensates exhibit subdiffusive behavior and move slower in confined states compared to non-confined.** **A** Mean MSD (μm²) ± s.d. of movement for non-confined (green) and confined (orange) condensates over 7 h. **B** Log fitting of mean MSD ± s.d. and time from (**A**). Condensates in confined and non-confined states have slopes 0.8 and 0.5, respectively, as depicted in illustrated lines. **C** Roseplot of centered condensate positional changes in *x*, *y* space. Each track represents condensate movements in non-confined and confined nuclei. **D** Boxplots of measured average absolute displacement (μm) per minute for HP1α condensates in non-confined (left box; *n* = 32 condensates [in 9 nuclei]) and confined (right box; *n* = 42 [in 10 nuclei]) in HeLa cells stably expressing HP1α-GFP (from 3 independent experiments). The boxplot represents the median (middle line), 25th, 75th percentile (box) and largest and smallest values extending no further than 1.5 × interquartile range (whiskers), except for outliers. ****: *p* ≤ 1.00e-04 from Mann–Whitney test two-sided.

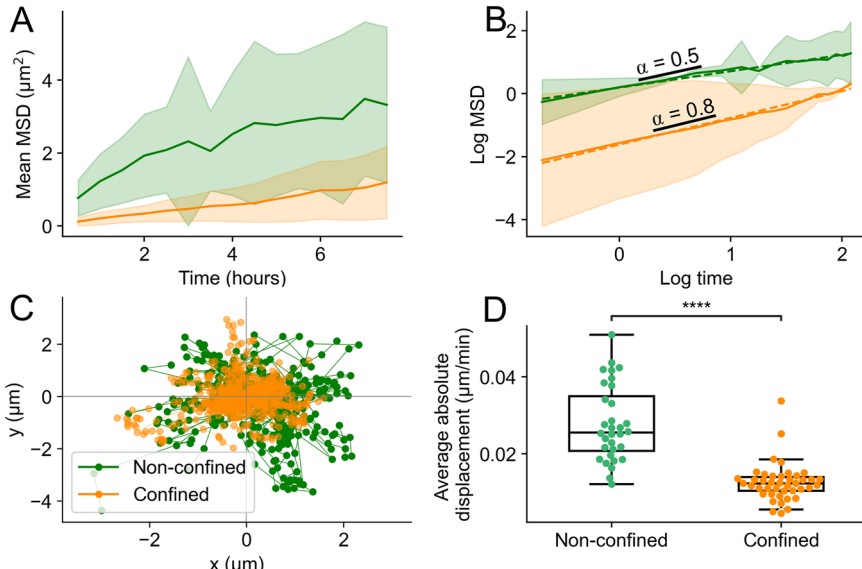

increased intranuclear density and/or viscosity from nuclear poroelasticity and decreased volume upon confinement, which is known to affect intranuclear dynamics[37]. Mechanical and osmotic stresses caused by e.g., nuclear flattening can also modulate nucleoplasmic viscosity and macromolecular crowding[38] potentially affecting the ability of HP1α condensates to diffuse and form. In addition, attractive interactions between HP1α condensates cause them to merge, minimizing their surface energy and bringing distant chromatin regions into close proximity to form transcriptionally silent foci[7,24]. These factors may contribute to explain the reduction in number and sizes of HP1α condensates, as well as their reduced dynamics. Due to their buffering properties, condensates seem less affected by HP1α concentration alterations induced by e.g., overexpression[39].

In our study, cells were subjected to confinement for 15 min which decreased their resting nucleus diameters to about half of their original heights. Even though many cells in the body, such as migrating immune cells, lung or skin cells, are subjected to transient deformations or compression, in other circumstances, as in cancer, compressive forces can persist over a long time[5,40–42]. Taking this into account, we tracked HP1α condensates of HeLa cells under confinement over time in order to study their dynamics. We found that HP1α condensates exhibit subdiffusive behavior with slope 0.5, which is plausible given obstructed motion within the nucleus (Fig. 4). This is also in agreement with previous findings reporting a slope for HP1α bound to heterochromatin below 1 quantified from fluorescence correlation spectroscopy (FCS) experiments[33]. In our study, we find that subdiffusive HP1α condensates are 4.5 times slower in confinement compared to non-confined states while covering significantly less total distance (Supplementary Fig. 6A). These results, taken together with the longer half-life confined condensates exhibit compared to non-confined (Supplementary Fig. 6B), indicate that HP1α condensates are more stable under confinement. Consistent with this, a study by Hsia et al. also finds HP1α condensates during confined migration to be more stable compared to non-confined migration[5]. While effects on HP1α condensates are observed upon immediate confinement, prolonged confinement suggests that condensates adopt stable patterns with less mobility. Over prolonged confinement for several hours, the stability of HP1α condensates might be influenced by replicative stress and cell viability.

Upon confinement, we observe HP1α condensates to centralize in the nucleus in both experimental studies and biophysical models on IMR90 (Figs. 6 and 7). A natural structure to compare with our observed HP1α condensates are senescence associated heterochromatin foci (SAHF) in IMR90 cells, a distinct heterochromatin structure associated with

H3K9me2,3 and HP1 proteins[43] observed during senescence. SAHFs are characterized by a redistribution of heterochromatin[43–45], detachment from the nuclear periphery[46] and relocalization of HP1 proteins and H3K9me3 to nuclear bodies[47]. Similarly, mechanical stretch and nuclear deformation activate Piezo1-mediated calcium release, reducing lamina-associated H3K9me3 heterochromatin and protecting the genome from mechanical stress and DNA damage[13]. In addition, nuclear deformation affects mechanosensing YAP/TAZ proteins, causing them to translocate to the nucleus[48], but they have also been shown to traffic to the cytoplasm in confined migration studies[49]. These factors could affect chromatin properties and the reorganization of HP1α condensates upon compression. Interestingly, this process is impaired in cancer cells, which may explain the lack of HP1α condensate centralization in our confined HeLa cells[13].

In summary, we observe that cell confinement and consequent nuclear flattening influence HP1α condensates in number, size, dynamics and localization. Our study sheds new light on how HP1α heterochromatin condensates respond to specific nuclear mechanoconfinement.

Given the important role of HP1 proteins in various cellular functions, including in heterochromatin formation[50], transcriptional regulation[51] and in maintenance of genome integrity[52], our results provide a foundational basis for future investigation into the important interplay between nuclear shape and chromatin in cell mechanoreception.

## Methods
### Cell culture
Stably transfected HeLa GFP:HP1α cells[53] (a kind gift of Dr. Paola Vagnarelli, Brunel University, London) were grown in Dulbecco's Modified Eagle Medium (DMEM; Lonza) supplemented with 10% FCS, 100 μ/ml penicillin and 100 μg/ml streptomycin with selection in G-418 (2 mg/ml) at 37 °C with 5% $CO_2$. IMR90 cells (ATCC) were grown in Eagle's Minimum Essential Media (EMEM; Lonza) supplemented with 10% FCS, 2 mM l-glutamine, 100 μ/ml penicillin, and 100 μg/ml streptomycin at 37 °C with 5% $CO_2$. Cells were routinely tested for mycoplasma.

### 6-well plate confiner
Confinement of IMR90 and HeLa was performed using a 6-well plate confiner (4DCell) similar to previous microconfinement methods described by Liu et al.[25] and Le Berre et al.[26] Large PDMS pistons were stuck on the coverlid. Coverslips with pillars from 2 to 5 μm fixed heights were put onto the pistons and when the coverlid is assembled, the pillars confine the cells including nuclei to the corresponding heights of the pillars. The PDMS

**Fig. 5 | FRAP analysis of HP1α-GFP indicates that HP1α exchange kinetics are slower in confined cells. A** HeLa cells stably expressing HP1α-GFP were imaged before (pre-bleached), during and after photobleaching of HP1α-GFP condensates (red circle) or of HP1α-GFP regions outside the condensates (yellow circle). Scale bar 5 μm. **B** Total linear fitted curves for bleached HP1α-GFP condensates with (orange, $N = 38$) and without (green, $N = 40$) confinement. **C** Total linear fitted curves for bleached areas outside HP1α-GFP condensates with (orange, $N = 37$) and without (green, $N = 41$) confinement. **D** $T_{1/2}$ recovery from the FRAP analysis (**B**). Values are mean ± s.d. of 38 independent experiments for HP1α-GFP condensates with confinement, and 40 independent experiments without confinement. Unpaired Student's $t$-test, ****$P < 0.0001$. **E** $T_{1/2}$ recovery from the FRAP analysis (**C**). Values are mean ± s.d. of 37 independent experiments for areas outside HP1α-GFP condensates with confinement, and 41 independent experiments without confinement. Unpaired Student's $t$-test, ****$P < 0.0001$.

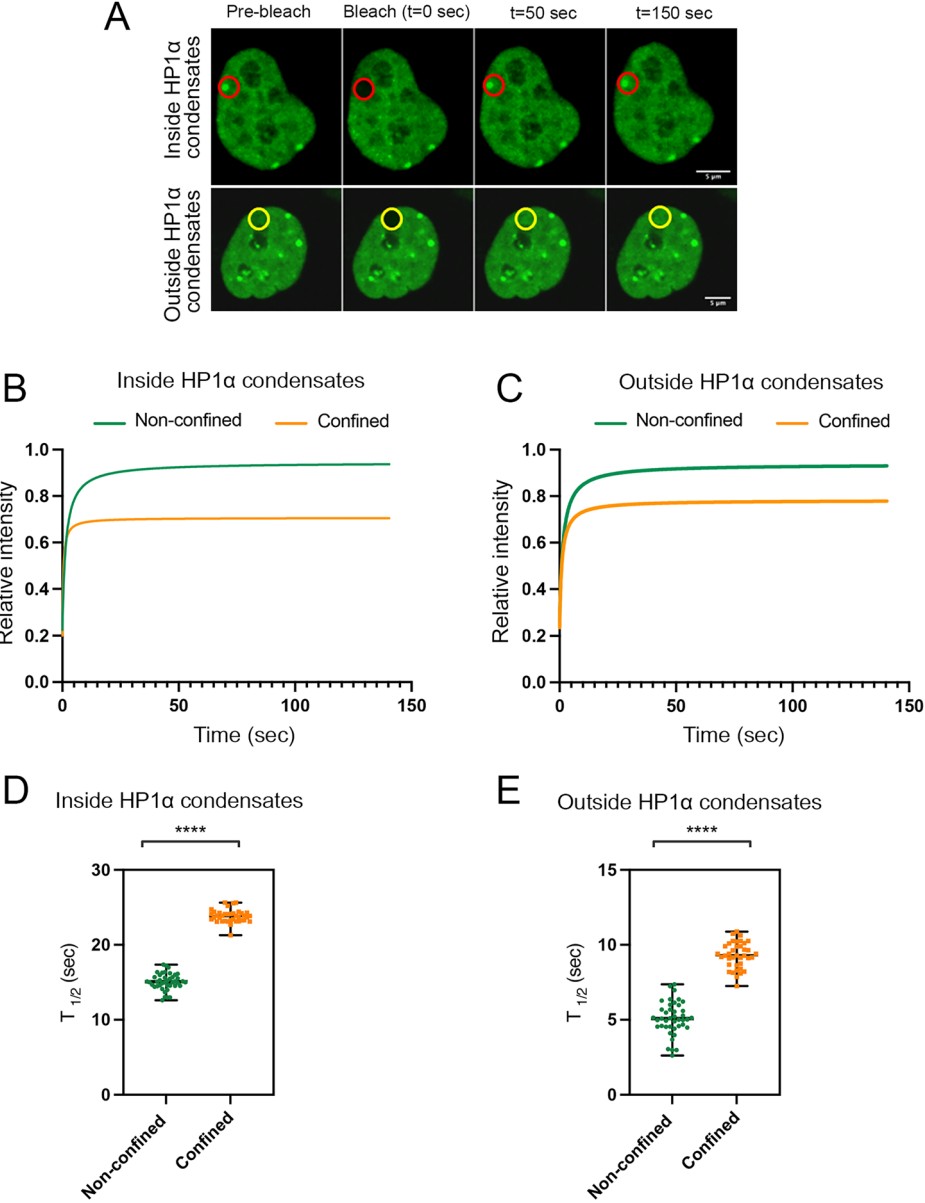

pistons, pillars and coverlid were cleaned in 70% ethanol and put under UV for 10 min before assembly. The coverlid is assembled by locking the metal handles into position in the holder which immediately puts a pressure around 20 kPa to the cells[27]. The cells and their nuclei were applied to confinement for about 15 min.

**Live cell imaging**
Cells were transfected by electroporation using Amaxa nucleofector 2b (Lonza) following the manufacturer's instructions. Around $1 \times 10^6$ cells were transfected with 2.5 μg of DNA using program X-001. Cells were subsequently plated in each of the wells in the 6-well 10 mm plate (MatTek). Nuclei were imaged the following day directly after confiner assembly on Zeiss LSM880 AiryScan Confocal at 37 °C and 5% $CO_2$. Z-stacks with 0.14 μm step size were generated with 63×/1.2NA oil objective with a resolution of 1024 × 1024 pixels and processed with the Zeiss Airyscan processing function (Wiener filtering). For nuclear staining, Hoechst 33342 dye (#639, Immunochemistry) was used at 1 μg/ml.

Confinement release experiments were conducted in HeLa cells stably overexpressing HP1α-GFP. Z-stacks with 0.2 μm step size were generated

with 63×/1.2NA oil objective with a resolution of 1000 × 1000 pixels. These Z-stacks were processed using Zeiss Airyscan processing. Z-stacks of nuclei were obtained under four different conditions: non-confined, immediate confinement (circa 5 min of confinement), after 15 min of confinement and after the confiner lid was released.

**Immunofluorescence microscopy**
Cells were fixed with methanol for 2 min at −20 °C, washed 3 times with 1× PBS and permeabilized with 0.5% Triton X-100 in 1× PBS for 15 min. After fixation and permeabilization, for cells subjected to confinement the confinement lid was removed and the staining was performed directly on the plate. Samples were incubated with primary antibodies diluted in the permeabilization solution for 20 min, washed 3 times with the permeabilization solution and incubated 20 min in secondary antibodies. After three additional washes, samples were incubated with DAPI 1:5000 in 1× PBS and washed with dH₂O. The samples were imaged in PBS using an Olympus SpinSR SoRA spinning disk confocal microscope equipped with a PLAPON 60×/1.42NA oil objective. Z-stacks with 0.24 μm step size were generated with a resolution of 2047 × 2035 pixels. The following primary antibodies were used for the IF staining: anti-HP1α (1:100, Sigma #05-689), anti-

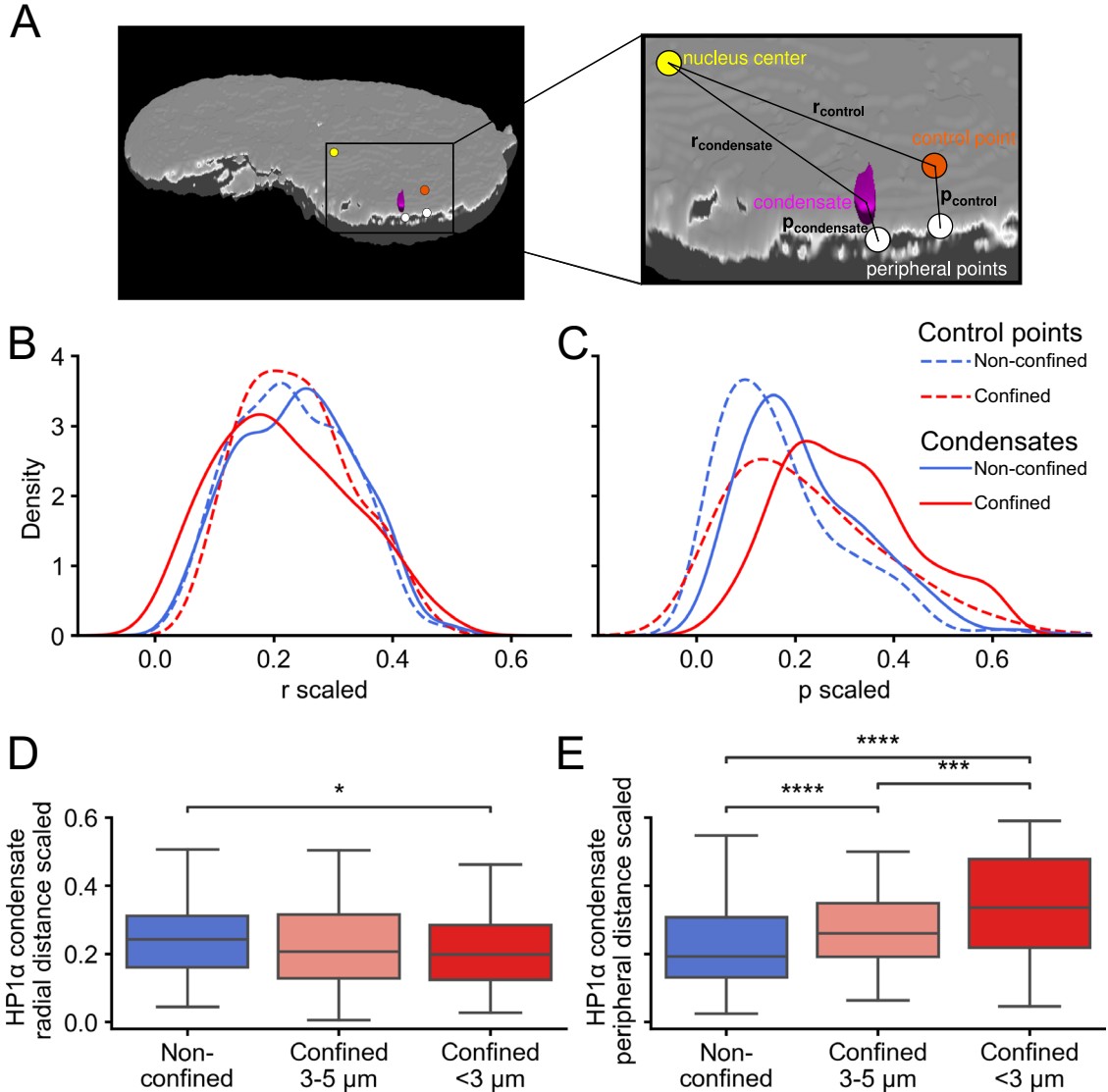

**Fig. 6 | HP1α condensates shift towards being localized in the center upon confinement. A** Illustration of radial and peripheral distances for a condensate (magenta) and a random control point (orange) in a tomographic view of a nucleus (gray). Radial distances (r) are measured from nucleus center (yellow) to center of condensate or control point scaled for nucleus diameter, while peripheral distances (p) from center of condensate or control point to nearest point on the nucleus periphery scaled for nucleus height (white points). **B** Kernel Density Estimation (KDE) plots of distributions of radial distances scaled for nucleus diameter of random points (control, dashed lines) and HP1α condensates (smooth lines) in non-confined (blue, 261 random points in control and 261 condensates) and confined cases (red, 181 random points in control and 181 condensates) for IMR90 cells.

**C** Same as (**B**), but showing nearest distance to the nuclear periphery scaled for nucleus height. **D** Boxplots of HP1α condensate radial distances scaled for nucleus diameter in IMR90 cells (left box; 261 condensates [in 19 nuclei] in non-confined, middle box; n = 98 [in 14 nuclei] in 3–5 μm and right box; n = 83 [in 13 nuclei] in <3 μm, from 6 independent experiments). **E** Same as (**D**), but showing nearest peripheral distances scaled for nucleus height. The boxplots in (**D** and **E**) indicate median (middle line), 25th, 75th percentile (box) and largest and smallest values extending no further than 1.5 × interquartile range (whiskers). *: 1.00e-02 < $p \le 5.00e-02$, ***: 1.00e-0 < $p \le 1.00e-0$, ****: $p \le 1.00e-04$ from Mann–Whitney test two-sided.

---

trimethyl-histone H3K9 (1:200, Sigma #07-442) and anti-LaminB1 (1:1000, abcam #ab16048). For nuclear staining, DAPI (Sigma-Aldrich) was used for fixed samples at 0.1 μg/ml.

**Image analysis**

Image processing and analysis were executed in Python version 3.11 and visualized in the 3D viewer Napari v.0.4.19[54]. Z-stack images were processed with a Gaussian blur filter and segmented by applying an intensity threshold to separate foreground and background voxels to generate all binary image. Nuclei were labeled using connected components labeling while condensates or strong chromatin regions were labeled with eroded-otsu labeling from py-clesperanto[55]. Quantifications for nuclei and HP1α condensates were extracted from HP1α-GFP isosurface objects. To verify

that nuclear measurements taken from the HP1α-GFP channel correspond to nuclei, a range of morphological and distance parameters were compared from the overexpressed signal to nuclear LaminB1 staining (Supplementary Fig. 9). The measurements show substantially high Pearson's correlation, thus justifying extracting nuclear parameters from the overexpressed staining.

To calculate nuclear elongation and flatness, we calculated major, intermediate and minor axes of the isosurface objects by performing a centralization of the physical coordinates followed by a rototranslation of all axes. This method retrieves bounding box measurements. We apply a finer estimation on the minor axis, the axis subjected to confinement, where 0.1% of voxels are excluded from top and bottom of the isosurface nuclei to remove irregularities.

**Fig. 7 | Localization of HP1α condensates within simulated shells. A** A schematic representation of human IMR90 H3K9me3 ChIP-seq data used to extract relative values of interaction strengths between chromosomal monomers. The monomers along the polymer are color-coded based on their interaction strengths, where dark red indicates high interaction strength and lighter colors represent lower strength. **B** showcases an example of an embedded polymer within a confining shell, assuming an oblate (squeezed) shape. **C–F** Illustrate the Kernel Density Estimation (KDE) distributions of normalized radial (R) shown in (**C** and **E**) and peripheral (P) values shown in (**D** and **F**) for HP1α as condensates. The distributions are presented for three confinement shapes (sphere, oblate 0.5, oblate 0.33), each at two different interaction strength values (E1 and E4).

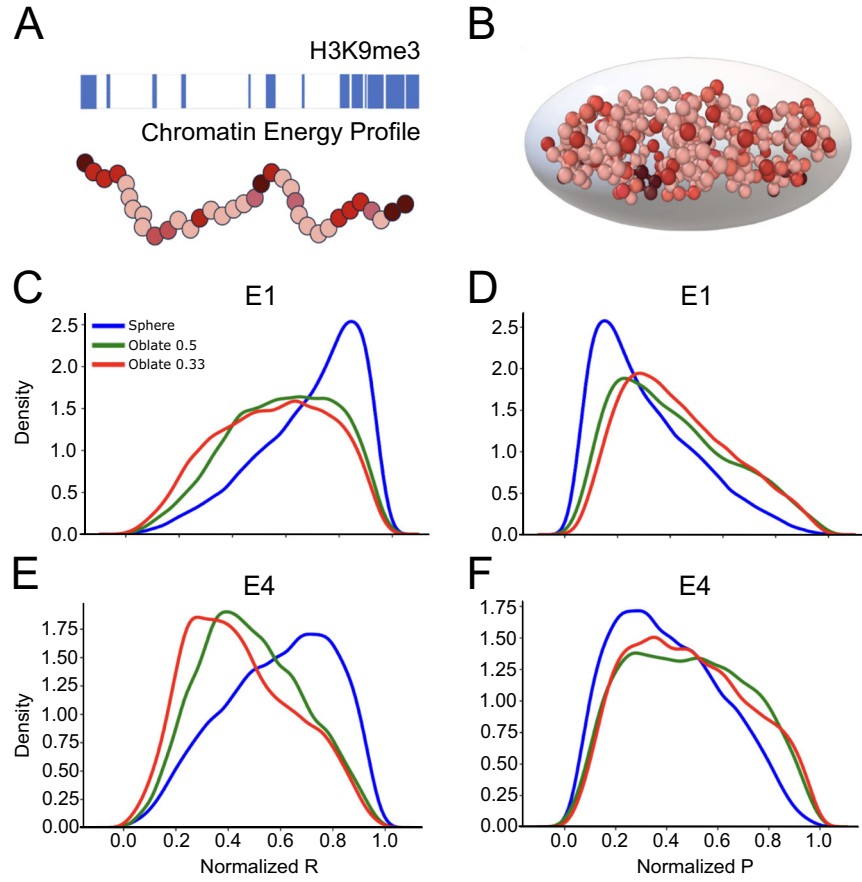

Nuclear elongation is calculated by

$$elongation = 1 - \sqrt{\frac{intermediate\ axis}{major\ axis}},$$

where 1 indicates the highest elongation.

Nuclear flatness i is calculated by

$$flatness = 1 - \sqrt{\frac{minor\ axis}{major\ axis}},$$

where flatness of 1 indicates a 1-dimensional flat plane.

Nuclear sphericity is calculated by

$$sphericity = \frac{\sqrt[3]{36\pi V^2}}{A},$$

where volume (V) and surface area (A) are calculated by multiplying the number of voxels making up the nuclei objects with the voxel and pixel sizes, respectively. A sphericity of 1 indicates a perfect sphere.

The center positions of the 3D nucleus and condensate objects were determined from their respective mean x, y, and z voxel coordinates. The radial distance was then calculated from the center of condensate to the center of the nucleus. The periphery of the nucleus was determined by taking the outer 0.05 fraction thickness shell of the whole nucleus object. Minimum distances were measured as Euclidean distances in μm from condensate centers to the closest voxel coordinate position on the periphery.

To account for varying sizes of HP1α condensates and their spatial localization within the nucleus, we defined volume regions based on the radius of each condensate. For each condensate, the radius distance was scaled by the nucleus diameter and an inner volume shell was defined by subtracting this fraction thickness from the whole nucleus object. Within each defined volume region, a single control point was randomly sampled for each condensate. Radial and peripheral distances were computed in the same manner as described for HP1α condensates above.

Colocalization of HP1α with H3K9me3 was quantified using Mander's colocalization coefficient, which measures the overlapping signal between the segmented objects from HP1α and H3K9me3 channels and divides it by the total signal in the HP1α channel.

**Tracking of HP1α condensates over time**

Live cell videos of nuclei from HeLa cell stably overexpressing HP1α-GFP were taken over a total timespan of 15 h. Images were acquired every 30 min with super-resolution in an Olympus SpinSR SoRA spinning disk confocal microscope equipped with a PLAPON 60×/1.42NA oil objective. Z-stacks with 0.5 μm step size were generated with a resolution of 2047 × 2035 pixels. A maximum intensity projection video was created and after applying a Gaussian blur filter, nuclei and condensates were segmented and labeled (see Image analysis). The tracking analysis was conducted using LapTrack[56] in Napari with an optimized track cost cutoff of 16,900 and splitting cost cutoff of 100 tuned by manual annotation. The HP1α condensate movements were extracted by correcting for nuclei movements at each time step by subtracting center nuclei x, y coordinates from center condensate x, y coordinates.

The mean square displacement (MSD) at time t is calculated by

$$MSD(t) = \frac{1}{N}\sum_{i=1}^{N}\left|\vec{r}^{(i)}(t) - \vec{r}^{(i)}(0)\right|^2,$$

where N is the number of condensates and $\vec{r}^{(i)}(0) = \vec{r}^{(i)}(t=0)$ the reference position of the i-th condensate. Mean MSD is calculated by taking the

mean of the MSD for all condensates in non-confined and confined states with standard deviations at each time point.

The anomalous diffusion exponent (α) is calculated from $x^2 = Dt^\alpha$ by taking the logarithmic transform $\log x^2 = \log D + \alpha \log t$, where the slope (α) describes the diffusion properties.

## FRAP

The FRAP experiments were performed on an inverted Olympus SpinSR SoRa spinning disk confocal microscope (Olympus, Hamburg, DE). The experiments were performed at 37 °C with 5% $CO_2$ with a PlanApo 60×/ 1.42 oil immersion objective. This spinning disk confocal microscope has two sCMOS Hamamatsu Orca Fusion cameras. The CellsSense software allows for continuous imaging of the GFP with the 488 nm laser and simultaneously bleaching with the 405 nm bleaching laser (maximum laser power for 800 ms). The click-and-bleach function in the cellSense software results in a dataset without post-bleaching lag time. Recovery was measured by acquiring images every 0,002101 s for 600 images. The obtained data were normalized and corrected for bleaching[57] and fitted by nonlinear regression to a function that assumes a single diffusion coefficient[58]:

$$F(t) = (F(0) + (F(\infty)((t/t1)/2)))/(1 + ((t/t1)/2))$$

The values for F0, F∞ and $t_{1/2}$ were calculated as described in Lippincott-Schwartz et al.[59] using GraphPad Prism 10.2.

## Polymer simulations

Polymer Model—We modeled sections of five randomly selected individual chromosomes with randomly selected p or q arm, namely chr8p, chr13q, chr14q, chr19p, and chr22q, as polymer chains. Each polymer chain was composed of spherical monomers representing consecutive 1 Mb sections of their respective chromosomes. These model chromosomes were confined in a shell representing the cell nucleus. In total, our model consisted of 258 monomers, equivalent to 258 Mbps in length.

To model the chromosomes, we employed the well-known Kremer and Grest polymer model[60]. Nearest-neighbor monomers along the contour of the individual chromosomes were connected by the finitely extensible nonlinear elastic (FENE) potential, given by:

$$U_{FENE}(r) = \{-0.5kR_0^2 \ln(1 - (r/R_0)^2), \text{if } r \leq R_0; \infty, \text{if } r > R_0\}$$

where $k = 30\varepsilon/\sigma^2$ is the spring constant and $R_0 = 1.5\sigma$ is the maximum extension of the elastic FENE bond. In order to maximize mutual polymer interpenetration at relatively moderate chain length and hence reduce the computational effort, we introduced an additional bending energy penalty between consecutive triplets of neighboring monomers along the polymer in order to control polymer stiffness:

$$U_{bend}(\theta) = k_\theta(1 - \cos(\theta))$$

Here, θ is the angle formed between adjacent bonds, and $k_\theta = 1\kappa BT$ is the bending constant. With this choice, the polymer is equivalent to a worm-like chain with Kuhn length $\ell_k = 2\sigma$[61].

The attraction between HP1α-bound segments of the chromosomes was simulated using the Lennard-Jones potential with a varied ε parameter. The interaction strength (ε) relies on the IMR90 H3K9me3 ChIP-seq data, where higher ε values were assigned to regions with stronger interactions.

As a direct mapping between the two interaction strengths (ChIP-seq data and simulations) is impossible, we considered their relative values and explored different levels of interaction strengths. Therefore, we conducted four sets of simulations using distinct interaction strength values (ε): low (E1), medium (E2), high (E3), and very high (E4). Within each set, the ratios of the interaction strength values between the segments were determined based on Chip-seq data, while only the absolute values varied.

Confinement Model—The confinement model involves the placement of N monomers on each selected geometry shape to create a confinement

shell. N represents the maximum number of equidistant monomers of diameter size σ that can be added on the desired surface. In this model, we considered three different confinement shapes: a spherical shape, an oblate shape flattened by a factor of two, and another oblate shape flattened by a factor of three. In all cases, the dynamics of all the monomer walls are set to zero, resulting in a rigid confinement. This rigid wall acts as a constraint for the polymers and models the nucleus, enabling the study of their behavior and properties within the confined space.

Density of the solution—We embedded polymers inside shells with fixed shapes. The sizes of these shells were chosen so that the fraction of the volume occupied by the monomers was approximately 10%. This volume fraction resembles the fraction of the nucleus occupied by chromatin[62], thus mimicking the packing and organization of chromatin inside the nucleus.

Measurement of condensates of HP1α-bound sites on chromosomes— Our aim was to quantify the condensates of HP1α-bound sites on the chromosomes through implicit HP1α simulations. As the number of clusters (condensates) is unknown beforehand, we used hierarchical agglomerative clustering to detect clusters/condensates in the simulation instances. The agglomerative clustering method employs a bottom-up strategy[63], initially assigning each monomer to its own individual cluster. Through iterative merging of the closest clusters, a dendrogram was constructed depicting the hierarchical relationships among all the monomers. To identify the optimal number of clusters, the dendrogram was cut at the distance corresponding to the highest silhouette coefficient. The silhouette coefficient serves as a measure of cluster quality, ensuring that the resulting clusters are cohesive and well-separated. This process effectively determined the optimal cutoff distance, leading to meaningful and well-defined clusters (condensates) of HP1α-bound segments.

## Input file for simulations

In order to model the interactions between HP1α proteins and chromosomes (polymers), we utilized Human IMR90 H3K9me3 ChIP-seq data (see Data availability) to extract relative values of the interaction strengths. These relative values were incorporated into the modeling process by determining the interaction parameter (ε) for the Lennard-Jones potential.

## Western blot

Cell lysis was performed by directly adding lysis buffer to the cells under confinement. Western blots were performed by loading 25–30 μg of total IMR90 cell lysates on 4–20% precast polyacrylamide gels (Bio-Rad). After transfer onto PVDF membrane, the blots were incubated overnight at 4 °C with the following primary antibodies: anti-HP1α (1:200, 05-689, Sigma/ Merck) and anti-GAPDH (1:2000, SC-47724, Santa-Cruz) for loading control. These were followed by incubation with species-matched HRP-conjugated secondary antibodies. The detection of the chemiluminescent signal was done using the ECL Prime Western Blotting Detection Reagents (Cytiva Amersham) and the ChemiDoc XRS+ System (Bio-Rad Laboratories). Densitometry analysis was performed using ImageJ software.

## Statistics and reproducibility

All boxplots represent the median, 25th and 75th percentiles (box) and whiskers extending to $1.5 \times IQR$, except for outliers. The numbers of independent experiments are stated in the figure legends, where 'n' values indicate the total number of HP1α condensates or chromatin foci analyzed from a specified number of nuclei. Statistical differences were assessed using a two-tailed Mann–Whitney test using the statannotations[64] Python package. In the figure legends, statistical significance is indicated as follows: *: $1.00e-02 < p \leq 5.00e-02$, **: $1.00e-03 < p \leq 1.00e-02$, ***: $1.00e-04 < p \leq 1.00e-03$, ****: $p \leq 1.00e-04$. For FRAP analysis, unpaired Student's t-test was used, and statistical significance is indicated as: ****$P < 0.0001$.

## Reporting summary

Further information on research design is available in the Nature Portfolio Reporting Summary linked to this article.

**Article**

## Data availability
The dataset for human IMR90 H3K9me3 ChIP-seq data used for polymer simulations can be found in the ENCODE database under the accession number ENCFFF625BTD.

## Code availability
The Python code used for analyzing the localization of HP1α condensates from polymer simulations is available here: https://doi.org/10.5281/zenodo.10853150[65]. Code for data analysis and source data tables for live-cell and tracking data are available at https://doi.org/10.5281/zenodo.14747054[66].

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

## Acknowledgements

We are grateful to Dr. Paola Vagnarelli (Brunel University, London) for the kind gift of the stably transfected HeLa GFP:HP1α cell line. We thank the NorMIC Oslo imaging platform (Department of Biosciences, University of Oslo) for the assistance with imaging and Sathiaruby Sivaganesh for technical support. We thank Dr. Andrea Raffo for assisting with the bounding box measurements for segmented objects from imaging. J.P. acknowledges funding from the Norwegian Research Council (project 324137). C.P. acknowledges the financial support from the Norwegian Cancer Society (project 223181), Astri og Birger Torsteds legat, and UNIFOR-FRIMED.

## Author contributions

J.P. and C.P. conceived the project. O.H carried out the imaging experiments and analyzed data. N.N. set up and performed polymer simulation models and analysis. O.H. and A.H. performed cell culture work and A.H. carried out the western blot experiments. L.H.H. performed the FRAP experiments and analysis. O.H., N.N., J.P., and C.P. wrote the manuscript with input from all authors. J.P. and C.P. procured funding.

## Competing interests

The authors declare no competing interests.
