## [Transparent Peer Review file · Communications Biology]

Nuclear mechano-confinement induces geometry-dependent HP1 α condensate alterations

Corresponding Author: Professor Cinzia Progidà

Version 0:

Reviewer comments:

Reviewer #1

(Remarks to the Author)

The manuscript submitted by Hovet and colleagues studies the influence of mechanical confinement on heterochromatin-associated HP1 α condensates, showing that upon squeezing of the cell nuclei to approximately half the height and volume, HP1 α condensates are partially lost, centralize in the cell and become less dynamic. In their revised manuscript, the technical issues raised by the reviewer in data interpretation and modeling are well addressed, methodological descriptions are clarified, and their findings overall appear robust and well validated. Although the functional implications and biophysical and molecular mechanisms involved in the observed alterations to HP1 α condensates remain mostly unaddressed in this work, I believe the current findings are of interest to the community, and I support publication without further revisions in Communications Biology.

Reviewer #2

(Remarks to the Author)

In their paper entitled "Nuclear mechano-confinement induces geometry-dependent HP1 α condensate alterations", the authors study HP1 α condensates in unconfined and confined cells, using two human cell lines (IMR90 and HeLa). They report that the numbers of HP1 α condensates decrease upon confinement, while the numbers of chromatin regions with intense DAPI staining remain the same. The diameter of HP1 α condensates also decreases upon confinement, and HP1 α becomes less mobile in FRAP experiments. The authors also observe a radial redistribution of HP1 α condensates upon confinement in IMR90 cells but not in HeLa cells.

The question how cells and chromatin react to mechanical stimuli is definitely an important and timely question, and the experiments presented in the manuscript appear of high quality and well-controlled. I think the main weakness is that the study remains rather descriptive. For a biology audience, it would be interesting to know if the alterations described here have consequences on biological functions (gene expression, signaling, genome stability, etc). The other way around, it would be interesting to know what causes these alterations, the mechanical constraints themselves or other processes triggered by confinement (mechano-signaling). Unfortunately, the authors do not provide many clues in this direction.

Some specific points that the authors might want to consider:

1. HP1 α condensates are typically considered to represent heterochromatin domains, which is also consistent with the colocalization with H3K9me3 described in this paper (Fig. 2, Fig. S3). How do the authors explain that the number of HP1 α condensates changes while the number of dense chromatin regions remains the same upon confinement (Fig. S5)? This finding is potentially interesting but poses many further questions: Does HP1 α relocate so that it dissociates from some dense chromatin regions but not from others? Which regions would this be? Mechanistically, what makes HP1 α leave, a change in histone marks? What distinguishes regions where HP1 α remains from those where it dissociates? I assume that experiments to address these questions are beyond the scope of the current paper, but it is very difficult to draw a biological conclusion without knowing the answers.

2. For a mechanistic understanding, it seems to be key to understand which alterations of HP1 α condensates are induced by geometrical constraints and which by signaling (e.g., nuclear import of YAP/TAZ, calcium signaling, change of histone

marks). I would encourage the authors to address this, either by monitoring signaling or by inhibiting signaling. As it stands, it is difficult to judge how realistic the simulated scenario is, which explains alterations with geometrical constraints. This is particularly an issue because some of the observed alterations are cell line-specific, i.e., the spatial redistribution that occurs in IMR90 but not in HeLa. Although I appreciate simple models, I want to add that many things that happen in the cell and could affect HP1 and heterochromatin are not captured in the simulation, for example attachment of chromatin to the lamina, attachment of chromatin to the nucleolus (which might pull heterochromatin that is close to ribosomal DNA loci towards the nuclear center), and biochemical signaling (change of histone marks upon confinement that will change the attachment of chromatin to the lamina and many other things).

3. The volume of confined HeLa cells (5-8 μm) does not seem to change (Fig. S2F), while that of IMR90 cells seems to change (Fig. S2C). According to Fig. 3, numbers and diameters of HP1 α condensates change under these conditions in both cell lines. Can the authors conclude anything from this observation? If HP1 α condensates are formed by LLPS, one could expect them to respond to a change in nuclear HP1 α concentration, which for 5-8 μm confinement would occur in IMR90 (same number of HP1 molecules in a smaller nuclear volume) but not in HeLa (same nuclear volume). If the alterations seen here do not depend on nuclear HP1 α concentrations, what do they depend on? Are they also geometry-induced, like the repositioning? Could this be reproduced with the polymer model? From a biophysics perspective, I am missing here a hint that could explain how the observed alterations in with respect to the number/size of HP1 α condensates (and of HP1 α dynamics) could be explained, using arguments from LLPS theory or from polymer simulations.

Reviewer #1 (Remarks to the Author):

The manuscript submitted by Hovet and colleagues studies the influence of mechanical confinement on heterochromatin-associated HP1a condensates, showing that upon squeezing of the cell nuclei to approximately half the height and volume, HP1a condensates are partially lost, centralize in the cell and become less dynamic. In their revised manuscript, the technical issues raised by the reviewer in data interpretation and modeling are well addressed, methodological descriptions are clarified, and their findings overall appear robust and well validated. Although the functional implications and biophysical and molecular mechanisms involved in the observed alterations to HP1a condensates remain mostly unaddressed in this work, I believe the current findings are of interest to the community, and I support publication without further revisions in Communications Biology.

Reviewer #2 (Remarks to the Author):

In their paper entitled "Nuclear mechano-confinement induces geometry-dependent HP1 α condensate alterations", the authors study HP1 α condensates in unconfined and confined cells, using two human cell lines (IMR90 and HeLa). They report that the numbers of HP1 α condensates decrease upon confinement, while the numbers of chromatin regions with intense DAPI staining remain the same. The diameter of HP1 α condensates also decreases upon confinement, and HP1 α becomes less mobile in FRAP experiments. The authors also observe a radial redistribution of HP1 α condensates upon confinement in IMR90 cells but not in HeLa cells.

The question how cells and chromatin react to mechanical stimuli is definitely an important and timely question, and the experiments presented in the manuscript appear of high quality and well-controlled. I think the main weakness is that the study remains rather descriptive. For a biology audience, it would be interesting to know if the alterations described here have consequences on biological functions (gene expression, signaling, genome stability, etc). The other way around, it would be interesting to know what causes these alterations, the mechanical constraints themselves or other processes triggered by confinement (mechano-signaling). Unfortunately, the authors do not provide many clues in this direction.

Some specific points that the authors might want to consider:

1. HP1 α condensates are typically considered to represent heterochromatin domains, which is also consistent with the colocalization with H3K9me3 described in this paper (Fig. 2, Fig. S3). How do the authors explain that the number of HP1 α condensates changes while the number of dense chromatin regions remains the same upon confinement (Fig. S5)? This

finding is potentially interesting but poses many further questions: Does HP1 α relocate so that it dissociates from some dense chromatin regions but not from others? Which regions would this be? Mechanistically, what makes HP1 α leave, a change in histone marks? What distinguishes regions where HP1 α remains from those where it dissociates? I assume that experiments to address these questions are beyond the scope of the current paper, but it is very difficult to draw a biological conclusion without knowing the answers.

As expected for heterochromatic domains, the percentage of colocalization between HP1 α and H3K9me3 is high (70% in IMR90 and 77% in HeLa, see Fig. 2E). The dense chromatin regions analysed in Fig.S5 correspond to regions stained by DAPI, which is not specific for HP1 α . Even if we expect HP1 α to be present in many of these high-density areas, DAPI staining cannot be used to identify HP1 α condensate regions. In line with this, it has been previously shown that the size and nuclear distribution of DAPI-stained pericentromeric heterochromatin does not change when HP1 proteins are displaced¹. We have included a sentence about this in the revised version of the manuscript (lines 431-433).

2. For a mechanistic understanding, it seems to be key to understand which alterations of HP1 α condensates are induced by geometrical constraints and which by signaling (e.g., nuclear import of YAP/TAZ, calcium signaling, change of histone marks). I would encourage the authors to address this, either by monitoring signaling or by inhibiting signaling. As it stands, it is difficult to judge how realistic the simulated scenario is, which explains alterations with geometrical constraints. This is particularly an issue because some of the observed alterations are cell line-specific, i.e., the spatial redistribution that occurs in IMR90 but not in HeLa. Although I appreciate simple models, I want to add that many things that happen in the cell and could affect HP1 and heterochromatin are not captured in the simulation, for example attachment of chromatin to the lamina, attachment of chromatin to the nucleolus (which might pull heterochromatin that is close to ribosomal DNA loci towards the nuclear center), and biochemical signaling (change of histone marks upon confinement that will change the attachment of chromatin to the lamina and many other things).

Our biophysical simulations capture heterochromatin properties under geometric constraints of a similar degree as the nuclei of IMR90 experience under confinement. We have shown that during this degree of confinement to IMR90 nuclei and from simulations, HP1 α condensates tend to centralize in the nucleus.

We have now added new modelling results showing that the number of HP1 α condensates is not altered upon nuclear flattening (Fig. S8), suggesting that the alteration in HP1 α condensates measured in our experiments is not just a consequence of geometrical constraints. Therefore, the observed reduction in HP1 α condensate number and size might be caused by multiple factors such as changed binding kinetics of HP1 α -HP1 α and HP1 α -DNA, DNA viscosity and forces applied to the condensates². The induced nuclear deformations will also affect signaling pathways such as nuclear import of YAP/TAZ³, Ca²⁺ nuclear influx^{4,5} and change of histone marks⁶. For instance, under strong nuclear deformations, YAP has been shown to translocate to

the nucleus³, however, confined migration studies show that YAP is trafficked to the cytoplasm⁷. Due to the complexity of the factors potentially involved, experiments to address these questions are beyond the scope of the current paper. Similarly, also setting up additional parameters to the simulations such as chromatin attachment to lamina and nucleolus would require a more complex and entirely new modelling project which is beyond the scope of this paper. We have now included some sentences in the discussion (lines 587-592 and 625-629) mentioning the possible pathways that could be involved in the alterations of HP1 α condensates.

3. The volume of confined HeLa cells (5-8 μm) does not seem to change (Fig. S2F), while that of IMR90 cells seems to change (Fig. S2C). According to Fig. 3, numbers and diameters of HP1 α condensates change under these conditions in both cell lines. Can the authors conclude anything from this observation? If HP1 α condensates are formed by LLPS, one could expect them to respond to a change in nuclear HP1 α concentration, which for 5-8 μm confinement would occur in IMR90 (same number of HP1 molecules in a smaller nuclear volume) but not in HeLa (same nuclear volume). If the alterations seen here do not depend on nuclear HP1 α concentrations, what do they depend on? Are they also geometry-induced, like the repositioning? Could this be reproduced with the polymer model? From a biophysics perspective, I am missing here a hint that could explain how the observed alterations in with respect to the number/size of HP1 α condensates (and of HP1 α dynamics) could be explained, using arguments from LLPS theory or from polymer simulations.

In Fig. S2C and S2F, the nuclear volumes of both HeLa and IMR90 cells do not show any significant change under moderate confinement (5-8 μm for HeLa and 3-5 μm for IMR90). Based on this, and also considering that HeLa cells are bigger than IMR90, and therefore we cannot directly compare the confinement height of these two cell lines, we do not agree with the reviewer's interpretation of a different response of HeLa and IMR90 nuclear volume to confinement.

Minor volume changes in HeLa nuclei under compression between 10 and 5 μm have previously been reported by Lomakin et.al, and explained by potential expansion of the nuclear envelope⁴. However, under high confinement (<5 μm for HeLa and <3 μm for IMR90), there is a substantial reduction in nuclear volume in both cell lines, consistent with micropipette aspiration experiments previously reported by Rowat et.al⁸. This has now been included in the text (lines 359-364).

Notably, at higher confinement levels, the changes in HP1 α condensate size, dynamics and repositioning become more evident. For HP1 α condensate numbers, there is a reduction in HeLa at higher confinement levels, while there is a minor increase in IMR90 between moderate and high confinement (Fig. 3B and 3F).

The reduction in HP1 α condensate numbers, sizes and dynamics might be explained from the limited space for HP1 α condensates to diffuse and form upon nucleus flattening. This involves repulsive and attractive interactions with LLPS and chromatin resulting in inhibited

condensate formation and chromatin rearrangement (previously discussed in lines 570-585 and now expanded with lines 586-592).

To answer the reviewer question, we have included new biophysical simulations (Fig. S8). Even though the new modelling shows no differences in total number of condensates between spherical and oblate conditions, suggesting that the alterations in the number of HP1 α condensates measured in our experiments is not simply geometry-induced, it cannot be excluded that geometric constraints might still partially affect the nuclei, in combination with other factors as for example signaling cascades, as now mentioned in the discussion (lines 625-629).

Added by the reviewer by email:

Regarding the specific points I had raised last time, the authors did not perturb HP1 (point #2), which I would have found helpful to establish “causality”, they did not figure out how condensate numbers go down (due to technical limitations, point #3), and they do not come up with a convincing model why the two cell lines are different (point #6). The additional experiment with DAPI (Fig. S5) actually confuses me a bit regarding the question if HP1 serves as a marker for heterochromatin domains (also point #2).

That being said, I generally agree with what the authors replied to my points (that it is not trivial to come up with a model for the difference between cell lines, that it is hard to maintain the focus when confining cells, etc), but it would have been nice to have some more insight into the mechanisms altering these condensates and to find out if the alterations “matter” (functionally).

We agree, and have added relevant discussion points in the Discussion section to address better which possible mechanisms could alter condensate properties (lines 587-592 and 625-629).

1. Mateos-Langerak, J. *et al.* Pericentromeric Heterochromatin Domains Are Maintained without Accumulation of HP1. *Mol. Biol. Cell* **18**, 1464–1471 (2007).
2. Keenen, M. M. *et al.* HP1 proteins compact DNA into mechanically and positionally stable phase separated domains. *eLife* **10**, e64563 (2021).
3. Elosegui-Artola, A. *et al.* Force Triggers YAP Nuclear Entry by Regulating Transport across Nuclear Pores. *Cell* **171**, 1397-1410.e14 (2017).
4. Lomakin, A. J. *et al.* The nucleus acts as a ruler tailoring cell responses to spatial constraints.

- Science* **370**, eaba2894 (2020).
5. Kalukula, Y., Stephens, A. D., Lammerding, J. & Gabriele, S. Mechanics and functional consequences of nuclear deformations. *Nat. Rev. Mol. Cell Biol.* **23**, 583–602 (2022).
 6. Hsia, C.-R. *et al.* Confined migration induces heterochromatin formation and alters chromatin accessibility. *iScience* **25**, (2022).
 7. Rianna, C., Radmacher, M. & Kumar, S. Direct evidence that tumor cells soften when navigating confined spaces. *Mol. Biol. Cell* **31**, 1726–1734 (2020).
 8. Rowat, A. C., Lammerding, J. & Ipsen, J. H. Mechanical Properties of the Cell Nucleus and the Effect of Emerin Deficiency. *Biophys. J.* **91**, 4649–4664 (2006).
 9. Banani, S. F., Lee, H. O., Hyman, A. A. & Rosen, M. K. Biomolecular condensates: organizers of cellular biochemistry. *Nat. Rev. Mol. Cell Biol.* **18**, 285–298 (2017).
 10. McCreery, K. P. *et al.* Mechano-osmotic signals control chromatin state and fate transitions in pluripotent stem cells. 2024.09.07.611779 Preprint at <https://doi.org/10.1101/2024.09.07.611779> (2024).
 11. André, A. A. M. & Spruijt, E. Liquid–Liquid Phase Separation in Crowded Environments. *Int. J. Mol. Sci.* **21**, 5908 (2020).